# Somatic Embryogenesis from the Leaf-Derived Calli of In Vitro Shoot-Regenerated Plantlets of *Rosa hybrida* ‘Carola’

**DOI:** 10.3390/plants13243553

**Published:** 2024-12-19

**Authors:** Mingao Duan, Juan Liu, Yining Zhao, Xiaofei Wang, Longzhen Li, Shiyi Wang, Ruidong Jia, Xin Zhao, Yaping Kou, Kairui Su, Hong Ge, Shuhua Yang

**Affiliations:** 1State Key Laboratory of Vegetable Biobreeding, Institute of Vegetables and Flowers, Chinese Academy of Agricultural Sciences, Beijing 100081, China; duanmingao@163.com (M.D.); liujuan3562856@163.com (J.L.); zyning10@126.com (Y.Z.); w17862688755@163.com (X.W.); m15877282678@163.com (L.L.); wangshiyi_sylvia@163.com (S.W.); jiaruidong@caas.cn (R.J.); zhaoxin01@caas.cn (X.Z.); kouyaping@caas.cn (Y.K.); gehong@caas.cn (H.G.); 2Key Laboratory of Biology and Genetic Improvement of Flower Crops (North China), Ministry of Agriculture and Rural Affairs, Beijing 100081, China; 3College of Horticulture and Gardening, Yangtze University, Jingzhou 434000, China; 4Shandong Hongmeng Talent Development Group Co., Ltd., Jinan 250400, China; 15553149888@163.com

**Keywords:** callus, plant growth regulator, plant regeneration, rose, somatic embryo

## Abstract

Roses are one of the most important flowers applied to landscape, cut flowers, fragrance and food industries widely. As an effective method for plant reproduction, the regeneration via somatic embryos is the most promising method for breed improvement and genetic transformation of woody plants. However, lower somatic embryogenesis (SE) induction rates and genotypic constraints impede progress in genetic transformation in rose. This study describes a plant regeneration system for the famous red cut flower cultivar *Rosa hybrida* ‘Carola’. The stems without petioles cultured on Murashige and Skoog (MS) medium supplemented with 1.0 mg·L^−1^ 6-benzylaminopurine (6-BA), 0.05 mg·L^−1^ a-naphthalene acetic acid (NAA) and 30 g·L^−1^ sucrose showed the maximum proliferation coefficient of shoots with 3.41 for the micropropagation system. We evaluated the effects of different plant growth regulators (PGRs) on the induction, proliferation and conversion of somatic embryos. The induction rate of calli reached 100% on MS medium supplemented with 2.0 g·L^−1^ NAA and 30 g·L^−1^ glucose. The highest induction rate of somatic embryos achieved a frequency of 13.33% on MS medium supplemented with 2.0 mg·L^−1^ zeatin (ZT), 0.1 mg·L^−1^ NAA and 30 g·L^−1^ glucose. The most suitable carbohydrate with 60 g·L^−1^ glucose resulted in a proliferation rate of somatic embryos (4.02) on MS medium containing 1.5 mg·L^−1^ ZT, 0.2 mg·L^−1^ NAA and 0.1 mg·L^−1^ gibberellic acid (GA_3_). The highest somatic embryos germination rate (43.33%) was obtained from the MS medium supplemented with 1.0 mg·L^−1^ 6-BA, 0.01 mg·L^−1^ IBA and 30 g·L^−1^ glucose. Finally, the germinated somatic embryos successfully rooted on 1/2 MS medium containing 1.0 mg·L^−1^ NAA, 30 g·L^−1^ sucrose, and the vigorous plantlets were obtained after hardening-off culture. This study provided a stable and efficient protocol for plant regeneration via somatic embryos in *R. hybrida* ‘Carola’, which will be beneficial to the further theoretical study and genetic improvement in roses.

## 1. Introduction

Rose is one of the most important ornamental plants throughout the world that has wide range of application and great economic value. Extensive crossbreeding of roses from China, Europe and the Middle East has formed the genetic basis for “modern rose cultivars” [1]. Nowadays, approximately 30,000–35,000 varieties of modern rose have been registered, but only 8–20 species have contributed to its origin among approximately 200 species in genus *Rosa* [2]. Some important traits have been lost in the process of continuous inbreeding, making rich flower colors and forms, fragrance, vase life, and resistance to pests and diseases to become the constantly sought after breeding goal [3,4,5]. However, the traditional breeding process is very slow and is restricted to some limiting factors, such as the high heterozygosity of genome, the lack of certain allelic variation and the long growth cycle of woody plants [6]. Meanwhile, some rose species or cultivars are triploid or diploid, while the majority of modern rose cultivars are tetraploid [7]. Therefore, it is very important to establish more efficient and reliable regeneration system for the different genotype of roses.

The plant regeneration patterns via tissue culture mainly include organ regeneration, somatic embryo regeneration, protoplast regeneration and protocorm-like body regeneration etc. [8,9]. Somatic embryogenesis (SE) was first reported in carrot cell suspension cultures [10,11], and now is applied to many plants [12,13,14,15,16]. Somatic embryos can be induced directly from the explant (direct SE) or induced after the formation of an intermediate callus stage (indirect SE) [6,17]. In Rosaceae family, both modes of genesis are present, but the indirect route is predominant [18]. Somatic embryos are the efficient recipients for genetic transformation since it is easy to isolate from the mother tissue due to the bipolar structure and independent vascular organization [19]. Moreover, genetic transformation via the single-cell originated somatic embryo can avoid the occurrence of chimera compared to the organogenesis [20]. The first report in roses of regeneration process was organogenesis pathway, the calli were induced from stem segments of hybrid tea rose ‘The Doctor’, and eventually obtained shoot primordia but no true shoots [21]. Valles and Boxus [22] first obtained the regenerated plants from somatic embryos in *R. hybrida*. Since then, there have been many studies on SE in the different rose species including *R. chinensis* [23], *R. rugosa* [24,25,26], *R. multiflora* [27], *R. canina* [28], and many varieties [5,6,29,30,31,32,33,34,35,36].

Many factors affect SE, including genotypes, explant sources, culture conditions, types and concentrations of plant growth regulators (PGRs), carbohydrates and so on. The induction of SE is strongly dependent on genotypes [17], with significant differences in induction rates among cultivars [29,31]. Yokoya et al. [37] verified that only 17 out of 24 rose varieties were induced to produce somatic embryos. Zakizadeh et al. [4] found that embryogenic callus (EC) could be induced in all 8 miniature rose cultivars except ‘Tiffany’ and ‘Andromeda’, moreover, ‘Sonja’ exhibited the highest somatic embryos induction rate but failed to convert into a plantlet. However, Dohm et al. [3] had proved that 69% roses of 50 tested genotypes could be regenerated using leaf explants, which developed an efficient regeneration protocol adapted to a broad range of rose genotypes. Somatic embryos can be obtained from various types of explants, such as leaf, petiole, stem, immature seed, petal, root, anther, zygotic embryo and protoplast, and leaves are considered to be the most commonly used and efficient explants for SE in roses [18]. Induction of EC and SE are usually cultured under dark conditions, but it was reported that EC can induce more embryos with expanded cotyledon (s) and reduce the frequency of abnormal shoots under red light [38]. SE can also be influenced by the components of the media, such as the type of culture media, carbon sources, the types and concentrations of PGRs. It is known that the basic culture media for induction of SE are Murashige and Skoog (MS), MS with half-strength macronutrients (1/2MS), Woody Plant Medium (WPM), Gamborg B5 Medium (B5), Schenk & Hildebrandt basal salt mixture (SH) and other categories. Endogenous auxin indole-3-acetic acid (IAA) and synthetic analog of auxin α-naphthalene acetic acid (NAA), 2,4-dichloropenoxyacetic acid (2,4-D) are offen used alone or in combination with cytokinins zeatin (ZT), kinetin (KT), 6-benzylaminopurine (6-BA) and cytokinin-like compounds thidiazuron (TDZ) to extensively promote plant regeneration through SE. 2,4,5-trichlorphenoxyacetic acid (2,4,5-T) is another synthetic analog of IAA, which was found to be very effective in the induction of somatic embryos. It was reported that 10 or 25 µM 2,4,5-T was the best for SE and was significantly more effective than 2,4-D in rose cultivar ‘Livin’ Easy’ [35], and 2,4,5-T could initiate regenerative callus as well as 2,4-D in rose cultivar ‘Cheweyesup’ [36]. Tissues and organs cultured in vitro are mostly heterotrophic that unable to synthesize their own carbohydrates, requiring additional carbon sources to maintain growth and development [39]. In addition, carbohydrates also play important role in maintaining osmotic potential [40]. Sucrose [23,38,41] and glucose [5,6] are commonly used for the induction of EC and somatic embryos in roses [27]. It was reported that the induction rate of EC was obviously higher in the media supplemented with glucose than sucrose [31], and the total callus production was close to 100% under glucose treatment for *R. hybrida* ‘Carefree Beauty’ [36].

Proliferation and germination of somatic embryos are both important for plant regeneration. Proliferation of somatic embryos is achieved through secondary SE, which has the advantages of time-saving and efficiency in plant regeneration, especially for woody plants which have long growth and reproduction cycle [5,34]. In roses, secondary SE was early observed by Rout et al. [42] and Hsia and Korban [31]. Li et al. [34] first reported the whole process of induction, proliferation, maturation and germination of secondary somatic embryos in roses. Abscisic acid (ABA), ethylene, and gibberellic acid (GA_3_) have important effects on the proliferation and maturation of somatic embryos [43,44,45]. ABA was found to be the most effective in promoting maturation and germination of somatic embryos [34]. Xing et al. [26] found that low light with 500–1000 lux remained somatic embryos were vigorous and yellow-golden, and obtained the highest proliferation coefficient with 4.00 in the medium containing 45 g·L^−1^ glucose in *R. rugosa* ‘Bao White’. The low germination rate in roses, coupled with the damage caused by *Agrobacterium*-mediated transformation, prevented the efficient acquisition of transgenic plants. In the germination process of somatic embryos, it has been found that the application of high-light density [33], low-temperature treatment [42], ABA [34], and 6-BA [46] can significantly increase the germination rate. Meanwhile, somatic embryos with two expanded cotyledon exhibited the higher germination and survival rates than those somatic embryos without expanded cotyledons, with one expanded cotyledon and polycotyledons in *R. chinensis* ‘Old Blush’ [23], suggested that the normality and development status of somatic embryos affected the subsequent germination and survival. Until now, there have been numerous studies on somatic embryogenesis of different species and cultivars in roses. However, it is still necessary to explore more efficient and reliable regeneration system via somatic embryogenesis due to the strong genotype dependence and lower transformation efficiency in roses.

Cut roses are often propagated by cuttings with great economic values. *R. hybrida* ‘Carola’ is a very popular cut rose cultivar with excellent ornamental and resistant traits that applied in China for decades. However, many of the cut roses including *R. hybrida* ‘Carola’ lack of perfume but full of prickles in the leaves and stems, which leaves the great potential for the genetic improvement of rose cultivars. In this study, an effective in vitro shoot regeneration protocol was firstly explored to obtain the sterile and uniform plantlets in *R. hybrida* ‘Carola’. Furthermore, the optimum culture conditions for the induction, proliferation and germination of somatic embryos were successively screened for providing the stable protocol for the indirect somatic embryogenesis of *R. hybrida* ‘Carola’. Our work aims to establish an efficient regeneration system via somatic embryogenesis in *R. hybrida* ‘Carola’ for genetic transformation that may contribute to the genetic improvement for roses.

## 2. Results

### 2.1. Establishment of the Micropropagation System

As shown in Appendix A, the two types of explant treatments, nodal segments with and without petioles, had little effect on the germination rate of explants, and removing the petiole did not cause major damage to axillary buds. However, the contamination rate was four times higher in the explants with petiole (26.67%) than without petiole (6.67%). It was mainly due to the fact that the petiole harbored pathogenic bacteria and increased the surface area of the explants, which led to incomplete disinfection. Therefore, we selected the explants without petioles to promote the initiation of axillary buds during the primary culture (Figure 1A).

The effect of different concentrations of 6-BA and NAA on the proliferation of axillary shoots were shown in Appendix A. In PGRs-free medium, the proliferation coefficient of shoots was the minimal and the young shoots were short with weak growth. When the concentration of 6-BA was 0.5 mg·L^−1^, the proliferation coefficient ranged from 1.70–1.96, which reduced the speed of propagation. The concentration of 1.5 mg·L^−1^ resulted in slight vitrification, which made it difficult to continue to be the material for the successional cultivation and expansion. On the other hand, when the concentration of NAA was 0.01 mg·L^−1^, the young buds were short, with an average height between 0.84–1.34 cm, slow growth and light green leaves. However, the concentration of NAA at 0.1 mg·L^−1^ was easy to produce calli at the basal incision, which impeded the proliferation of shoots, and even led to serious withering. When the concentration of 6-BA was 1.0 mg·L^−1^ and the concentration of NAA was 0.05 mg·L^−1^, the proliferation coefficient of shoots reached 3.41 with the robust growth, thick and sturdy stem, dark-green leaf color (Figure 1B). Therefore, it was the most suitable medium for the proliferation of *R. hybrida* ‘Carola’.

The effects of different concentrations of NAA on the rooting varied significantly (Appendix A). The average length of roots was around 0.98–1.44 cm when the concentration of NAA was lower than 0.05 mg·L^−1^, but the average number of roots was less between 4.67–5.17. It had a significant inhibitory effect on the root elongation when NAA was higher than 0.1 mg·L^−1^. The average number of roots showed an increasing trend, as the concentration of NAA continued to increase to 0.5 mg·L^−1^, reaching a maximum of 11.85. But the average root length was around 0.36–0.86 cm, extremely short and weak. The NAA concentration of 0.05 mg·L^−1^ was chosen as the best rooting medium for *R. hybrida* ‘Carola’, and the average length of roots was around 2.52 cm. The roots were thick, strong, fast-growing and well-developed showing the best state of growth (Figure 1C).

Hardening-off time affect the survival rate of tissue culture plantlets (Appendix A). After 6 days of hardening-off treatment, 94.44% of the plantlets grew new leaves on the top, which had strong growth, dark green and shiny leaves (Figure 1D).

### 2.2. The Induction of Calli

The effect of NAA and different kinds of cytokinins on callus induction were shown in Table 1. The callus induction rate in different cytokinins with 2.0 mg·L^−1^ NAA all reached 100% when the explants were unexpanded leaves of sterile plantlets. However, the growth status of the calli varied significantly depending on the media supplemented with different concentrations of cytokinins. All calli formed in the darkness would be soft and yellow in color. The soft and creamy-white calli began to emerge arose from the margins of the leaves on Callus Induction Medium (CIM). It was observed that the best growth of calli were obtained on MS medium without the addition of 1.0 mg·L^−1^ ZT, KT, 6-BA with a crispy texture, bright color and faster growth rate (Figure 2A). When one of the three cytokinins was added, although the induction rate of the callus was also 100%, the growth was slow, the proliferation effect was poor, and browning and death were easy to occur in the later successional culture. Therefore, the optimal CIM was A1: MS + 2.0 g·L^−1^ NAA + 30 g·L^−1^ glucose + 2.5 g·L^−1^ phytagel.

### 2.3. The Induction of Somatic Embryos

The formation of EC is essential for SE, and the calli were transferred to Embryo Induction Medium (EIM) which can undergo embryogenic process to produce EC and then gives rise to somatic embryos. The effect of different concentrations of TDZ, NAA and GA_3_ on SE was shown in Table 2. Significant differences were observed in the induction of SE under different PGRs combinations, with the highest induction rate of 10% in the medium supplemented with 1.0 mg·L^−1^ TDZ, 0.5 mg·L^−1^ NAA and 1.0 mg·L^−1^ GA_3_. Comparison of B5 and B9 media showed that the addition of 1.0 mg·L^−1^ GA_3_ significantly increased the induction rate of somatic embryos. Among the 12 combinations, the induction rate of somatic embryos was 0 when the concentration of TDZ reached 3.0 mg·L^−1^ and above adding the same concentration of NAA and GA_3_, indicated that too high a concentration of TDZ (3.0 mg·L^−1^ and above) inhibited SE. The concentration of TDZ between 1.0–2.0 mg·L^−1^ was more appropriate than others.

Then we further explored the greater induction frequency of SE. There were significant differences in the induction rates of somatic embryos treated with different combinations of concentrations of ZT and NAA (Table 3). The highest induction rate of somatic embryos, 13.33%, was obtained on MS medium supplemented with 2.0 mg·L^−1^ ZT and 0.1 mg·L^−1^ NAA. This rate was significantly higher than that obtained in B2 medium, which was supplemented with 2.0 mg·L^−1^ TDZ and 0.1 mg·L^−1^ NAA, proving that ZT was more effective than TDZ in inducing somatic embryos. At this stage, white and translucent primary somatic embryos grew on the surface of the EC (Figure 2B). Therefore, the optimal EIM for *R. hybrida* ‘Carola’ was C2: MS + 2.0 mg·L^−1^ ZT + 0.1 mg·L^−1^ NAA + 30 g·L^−1^ glucose + 2.5 g·L^−1^ phytagel.

### 2.4. The Proliferation of Somatic Embryos

After culturing for up to 8 weeks on EIM, the EC were placed on Embryo Proliferation Medium (EPM) to proliferate with different concentrations of carbohydrate and PGRs (Table 4). We found the nodular, creamy yellow, translucent secondary somatic embryos grew from the surface of the primary somatic embryos after 4 weeks of culture (Figure 2C). The initial embryogenic cultures were not synchronous so that various stages of somatic embryos, including globular, torpedo, heart-shaped and cotyledonary stage were observed (Figure 2D–G). The proliferation coefficient of somatic embryos reached 3.65 in D2 and 4.02 in D4, both media containing 60 g·L^−1^ glucose, and these were higher than those in D1 and D3 media, which contained 30 g·L^−1^ glucose, with good glossiness of the somatic embryos. It can be seen that the addition of 1.5 mg·L^−1^ TDZ may produce abnormal embryos with a single cotyledon, several cotyledons or fused cotyledons, while the addition of 1.5 mg·L^−1^ ZT did not produce abnormal embryos and somatic embryos grew vigorously. Therefore, the optimum EPM for *R. hybrida* ‘Carola’ was D4: MS + 1.5 mg·L^−1^ ZT + 0.2 mg·L^−1^ NAA + 0.1 mg·L^−1^ GA_3_ + 60 g·L^−1^ glucose + 2.5 g·L^−1^ phytagel.

After 4 weeks of cultivation, the somatic embryos grew under light showed a markedly superior growth status to those in the dark. Under light condition, the somatic embryos exhibited rapid proliferation and good luster. However, the somatic embryos displayed slow proliferation and even browning or blackening. Consequently, it is advisable to select light conditions for the propagation and preservation of somatic embryos in *R. hybrida* ‘Carola’.

### 2.5. The Conversion of Somatic Embryos

When most of the somatic embryos turned to mature cotyledonary somatic embryos, we transferred them in Embryo Germination Medium (EGM) containing 1.5 mg·L^−1^ TDZ with different kinds of 6-BA, KT, GA_3_ and ABA (Table 5). Comparing the results on E1 and E4 media, the highest germination rate of 17.72% was achieved when somatic embryos were grown on medium containing 1.5 mg·L^−1^ TDZ with an additional 0.5 mg·L^−1^ 6-BA, which was significantly better than KT. Supplementation of GA_3_ and ABA had no effect on germination rate of somatic embryos. Moreover, GA_3_ caused the somatic embryos to turn green, harden, and die easily by browning in subsequent culture. ABA did not promote the germination of somatic embryos but promoted the maturation of somatic embryos, slow down the proliferation and produce abnormal embryos that eventually fails to germinate during subculture.

Based on the above results, 6-BA was selected as the effective cytokinin component to continue the study with different kinds of auxins (Table 6). Under light conditions, somatic embryos gradually turned green and produce shoots that gradually extend (Figure 2H,I). Using only 6-BA did not significantly enhance the germination of somatic embryos into plantlets, and when the concentration of 6-BA was increased to 1.0 mg·L^−1^, all combinations of somatic embryos germinated. The results on F3, F4, and F5 media showed that on the medium with the combination of the 6-BA and indole-butyric acid (IBA), the germination rate and plantlets number of somatic embryos were the highest at 43.33% and 1.31, which was significantly better than growing with NAA (10.34%, 0) or 2,4-D (31.03%, 0.89). Therefore, the most optimum EGM for *R. hybrida* ‘Carola’ was F4: MS + 1.0 mg·L^−1^ 6-BA + 0.01 mg·L^−1^ IBA + 30 g·L^−1^ glucose + 2.5 g·L^−1^ phytagel.

The shoots from germinated somatic embryos were individually separated and successfully rooted on Rooting Medium (RM): 1/2 MS + 1.0 mg·L^−1^ NAA + 30 mg·L^−1^ sucrose + 6.5 mg·L^−1^ carrageenan cultured under light conditions after about 4 weeks of culture (Figure 2J,K). The rooted plantlets gradually developed new leaves after 6 days hardening-off culture in greenhouse and grew well (Figure 2L).

## 3. Discussion

Roses are predominantly propagated by cutting with branches or grafting onto rootstocks. Micropropagation can shorten production time and produce healthy, disease-free roses to increase yields. Martin [47] proposed that this technique enables the propagation of up to 400,000 individual plants annually, all derived from a single rose. As we known, numerous rose cultivars successfully built the in vitro micropropagation systems [4,5,36,38,39,48,49,50,51]. In this study, an effective micropropagation system for *R. hybrida* ‘Carola’ with a proliferation coefficient of 3.41 were established under MS medium containing 1.0 mg·L^−1^ 6-BA, 0.05 mg·L^−1^ NAA, 30 g·L^−1^ sucrose, and 6.5 g·L^−1^ carrageenan. Compared with most of the previous studies [49,52], we obtained the high proliferation multiplication with almost the lowest concentration of PGRs combination, which provides a practical and cost-saving protocol for rose in vitro regeneration. In addition, just a few wild species such as *R. laevigata*, *R. canina* [53], *R. wichurana* [39] and *R. damascena* [54,55] could be propagated by micropropagation, which remains us to further disclose the genotype-depended regeneration mechanism.

This is the first report of SE in the popular cut rose cultivar *R. hybrida* ‘Carola’. In this study, the unexpanded compound leaves of sterile plantlets are applied as explants, since leaves are the most commonly used explant material in the induction of rose callus. The leaflets of *R. hybrida* ‘Carola’ cultured on MS medium supplemented with 2.0 mg·L^−1^ NAA alone and combined with 1.0 mg·L^−1^ ZT, KT or 6-BA all exhibited 100% induction rate for the calli appearance (Table 1), indicating that the leaflets of *R. hybrida* ‘Carola’ may be much easier to induce the callus than other genotypes. However, the combined PGRs made the calli grow slowly, suggesting that the application of single auxin is sufficient and more effective than the PGRs combinations to induce calli of *R. hybrida* ‘Carola’ (Figure 2A). Similar to our results, Vergne et al. [56] found 50% explants of *R. chinensis* ‘Old Blush’ developed whitish calli at the incision sites in CIM only containing four concentrations of NAA, and the larger calli were produced at the concentrations of 8.06 and 10.74 µM NAA than 2.69 and 5.37 µM NAA.

EIM was used to induce EC and primary somatic embryos. Calli are divided into EC and non-embryogenic callus (NEC), and the EC can be maintained for a long time without losing the ability to develop into somatic embryos through subculture [25]. The description of the EC and NEC varied in different rose species. Rout et al. [42] found the EC were light green and highly globular, NEC were white and friable in *R. hybrida* ‘Landora’. However, the leaflets of *R. hybrida* ‘Carefree Beauty’ could be induced with three kinds of calli: embryonic, organogenic and non-differentiated calli [34]. Cai et al. [23] identified white and reddish-brown translucent calli were the main types of EC to be induced in *R. chinensis* ‘Old Blush’. Likewise, we found the translucent and granular calli with appearance of somatic embryos began to show embryogenicity (Figure 2B).

As we known, SE is highly depended on the genotype and there was no universal protocol applied to the induction of somatic embryos in different roses till now [34,57]. Auxin plays a crucial role for somatic embryo formation [58]. 2,4-D is considered as the most effective auxin for the induction of EC in many rose cultivars [6,23,25]. However, SE can also be induced by high concentrations of auxin analogs such as NAA. Zakizadeh [4] did not observed EC formation across various concentrations of 2,4-D. Similar to this result, Vergne et al. [56] found that if replaced NAA with 2,4-D, these experiments failed to yielded EC in *R. chinensis* ‘Old Blush’, suggesting that NAA might be better than 2,4-D in the induction of EC in the certain genotypes. Different concentrations of NAA in combination with cytokinins can effectively promote SE in rose cultivars [3,42,59]. We obtained the highest induction rate with 13.3% of somatic embryos in the C2 medium supplemented with 2.0 mg·L^−1^ ZT and 0.1 mg·L^−1^ NAA, but the remaining combinations did not yield any somatic embryo (Table 3). Noriega and Söndahl [59] found that when the ratio of NAA to ZT is relatively low (0.25/1.5), it may in favor of the friable embryogenic tissue induction in rose cultivar ‘Royalty’. Kim [60] found the induction rate of SE for *R. hybrida* ‘4th of July’ reached 17.7% when the ratio of NAA/ZT was 5.5/18.2. In our research, the maximum somatic embryos induction rate with 13.3% was obtained when the NAA/ZT ratio was 0.1/2. The B9 medium with the addition of 1.0 mg·L^−1^ GA_3_ containing 1.0 mg·L^−1^ TDZ and 1.0 mg·L^−1^ NAA resulted in a significant increase of somatic embryos induction (10.00%) than in B5 medium (3.33%) without GA_3_ in *R. hybrida* ‘Carola’ (Table 2). It is known that there was the crosstalk between gibberellin and auxin signaling pathways, referring that GA_3_ may collaborate with NAA to improve the occurrence of SE. In *R. hybrida* ‘Carefree Beauty’, EC was observed with the medium containing 9.1 µmol·L^−1^ TDZ and 2.9 µmol·L^−1^ GA_3_, whereas it was absent in the media with the elevated TDZ concentrations [34]. Zakizadeh [4] found that 6 miniature rose cultivars could generated EC on the medium containing different concentrations TDZ, but ZT induced EC only in 3 cultivars. In our study, there were no somatic embryo in the media once the concentration of TDZ over 3.0 mg·L^−1^ (Table 2), indicating that the high concentration of TDZ may inhibit SE of roses. The combination of 2,4-D and TDZ generated more globular somatic embryos from EC, but TDZ alone with high concentrations led to the abnormal somatic embryo in *R. chinensis* ‘Yueyuehong’, which appeared “wood ear”-like plants with simple leaves, then finally browned and died in 11.25 µM TDZ [38]. The suitable endogenous auxin level is crucial for the formation and maintenance of somatic embryo, since auxin is the pivotal hormone that may evoke the totipotency-related genes and early embryogenesis genes to initiate SE [58]. Unlike the above results, the leaflets of cut rose *R. hybrida* ‘Ocen song’ did not present EC with different concentrations of NAA (1.0, 2.0, 3.0 and 4.0 mg·L^−1^) alone, but exhibited the highest frequency of EC (46.66%) and primary embryos (7.33%) on MS medium supplemented with 3.0 mg·L^−1^ NAA in combination 300 mg·L^−1^ proline [61]. It is possible that proline as an important osmotic regulator can help the cells of SE against the osmotic stress during the tissue culture [62].

The frequency of SE is generally low and exhibits the significant cultivar-depended variability, usually between approximately 3%~30% [25,63]. Thus, it is very important to obtain the secondary somatic embryos with the easy and simple process after the induction of primary somatic embryos [5,31,56]. The proliferation and germination of secondary somatic embryos have been widely reported in various rose species [5,23,26,56]. Furthermore, secondary somatic embryos have a high multiplication rate and uniformity with an independence of the original explants quantity [26]. At this phase, light conditions, carbon sources and PGRs concentrations have great influence on the proliferation and growth status of somatic embryos. Our study found that somatic embryos for *R. hybrida* ‘Carola’ presented significantly better growth state with light treatment than those under darkness. In the light, somatic embryos were of golden-yellow color, fast growth rate and high proliferation. In contrast, somatic embryos cultured under the dark conditions were weak, waterlogged, non-proliferating and even gradually browning to death. Similar studies on the light induced occurrence of secondary somatic embryos were found in *R. hybrida* ‘Samantha’ [5] and *R. rugosa* ‘Bao White’ [26]. The high concentrations of carbohydrate that sustained the suitable osmotic pressure could promote SE [64]. In our experiment, all of the media containing 30 and 60 g·L^−1^ glucose promoted the proliferation of secondary somatic embryos (Figure 2C). Bao et al. [5] studied the effect of carbohydrate type and concentration on the secondary somatic embryos of *R. hybrida* ‘Samantha’. The higher proliferation coefficient and lower percentage of abnormal appearance were observed in the secondary somatic embryos cultured in 30 g·L^−1^ (4.4, 30%) and 60 g·L^−1^ (3.8, 15%) glucose than those in 30 g·L^−1^ (2.4, 50%) and 60 g·L^−1^ (3.2, 50%) sucrose. Moreover, there were compact and golden-yellow secondary somatic embryos when glucose as the only carbon source. Xing et al. [26] further found the healthy and golden-yellow secondary somatic embryos appeared on the medium containing 45 g·L^−1^ glucose with the highest proliferation coefficient (4.00) as compared to those media containing the glucose concentration of 30 g·L^−1^ (3.48) or 60 g·L^−1^ (3.20), indicating that the suitable glucose concentration is vital for the proliferation of secondary somatic embryos. Kim et al. [60] used MS medium containing 30 g·L^−1^ sucrose supplemented with 6.8 µM ZT, 0.5 µM NAA and 2.9 µM GA_3_ to encourage the proliferation of EC in *R. hybrida* and *R. multiflora*. Vergne et al. [56] transferred EC to the MS medium containing 30 g·L^−1^ sucrose, 1.34 µM ZT, 6.84 µM NAA and 3 µM GA_3_ for proliferation in *R. hybrida* ‘Old Blush’. In our study, the secondary somatic embryo with the maximum proliferation coefficient of 4.02 was obtained on D4 medium with a glucose concentration of 60 g·L^−1^ supplemented with 1.5 mg·L^−1^ ZT, 0.2 mg·L^−1^ NAA and 0.1 mg·L^−1^ GA_3_. Moreover, there were better glossiness and lower abnormal appearance in secondary somatic embryos cultured in D4 medium (Table 4). Although the abnormal somatic embryos frequently appeared during proliferation in roses, Bao et al. [5] surprisingly found there were less abnormal somatic embryos cultured on MS medium containing 30 or 60 g·L^−1^ glucose than on sucrose for *R. hybrida* ’Samantha’. In this study, the morphologically abnormal somatic embryos on D1 medium containing 1.5 mg·L^−1^ TDZ, 0.2 mg·L^−1^ NAA and 0.1 mg·L^−1^ GA_3_, but the normal and better growth state on D3 medium containing 1.5 mg·L^−1^ ZT, 0.2 mg·L^−1^ NAA and 0.1 mg·L^−1^ GA_3_. Obviously, ZT should be more suitable than TDZ to avoid the occurrence of abnormal embryos during the proliferation process of somatic embryos in *R. hybrida* ‘Carola’.

Germination of somatic embryos are crucial for the establishment of a regeneration system for the SE pathway. GA_3_ may present a significant positive effect during somatic embryos growth and development in roses [65]. The percentage of embryo regeneration to shoots in *R. hybrida* ‘Linda’ was significantly increased from 60% to 70% when the concentrations of GA_3_ increased from 0.29 µM to 1.44 µM in the medium containing 1.42 µM NAA and 17.76 µM 6-BA [4]. Nonetheless, GA_3_ had no effect on the regeneration, but led to the abnormal development or premature germination of somatic embryos [33]. In our study, GA_3_ may have a significant inhibitory effect on somatic embryo germination and plant regeneration in *R. hybrida* ‘Carola’. As shown in Table 5, the presence of GA_3_ on E2 medium severely reduced the germination rate and the number of regenerated plantlets, and led to green-color and hardened somatic embryos which eventually died of browning in the subsequent culture. Similarly, ABA effects on germination of somatic embryo varied with the genotypes. There were five times higher germination rate for *R. hybrida* ‘Carefree Beauty’ on the medium with ABA than those cultured on the medium containing 6-BA and TDZ [34]. In contrast, ABA was implicated as an inhibitor of plant regeneration in floribunda rose cultivars *R. hybrida* ‘Trumpeter’ and ‘Glad Tidings’ [57]. In this study, the addition of ABA in the medium resulted in abnormal embryos and significant decreases in the germination rate of somatic embryos, which ultimately no any survived plantlet in *R. hybrida* ‘Carola’. 6-BA was widely used for somatic embryo germination in roses [5,6,26,38,62,63]. The germination rate of somatic embryos under the medium containing 1.0 mg·L^−1^ 6-BA alone was 63% in *R. rugosa* ‘Bao White’ [26] but was only 6.9% in *R. hybrida* ‘Carola’ (Table 6), suggesting the genotype depended efficiency of 6-BA on plant germination in roses. To improve the germination rate, 6-BA combined with NAA, IBA and 2,4-D were applied to the medium for *R. hybrida* ‘Carola’. The highest germination rate with 43.33% was obtained on F4 medium containing 1.0 mg·L^−1^ 6-BA and 0.01 mg·L^−1^ IBA, indicating that IBA could be the better auxin than NAA or 2,4-D to combined with 6-BA during somatic embryo germination process of *R. hybrida* ‘Carola’ (Table 6) and other cultivars [32,57,60].

During the past decades, the molecular mechanism of SE has been intensively studied in Arabidopsis [66]. As mentioned before, auxin is the pivotal hormone that initially reprograms somatic cells into a totipotent state by modulating transcription factors (TFs) such as WUSCHEL (WUS), BABYBOOM (BBM), LEAFY COTYLEDON (LEC), and finally contributes to somatic embryo formation by evoking early embryogenesis TFs like WUSCHEL-RELATED HOMEOBOX (WOX2/3) [66]. *BBM* was rapidly upregulated in explants under 2,4-D treatment, and ectopic expression of *BBM* developed somatic embryo-like structures on the edges of cotyledons and leaves in Arabidopsis and rice [67,68]. The *wox2 wox3* double mutation significantly inhibited SE in Arabidopsis, but transient overexpression of either *WOX2* or *WOX3* alone did not induce somatic embryo in hormone-free MS media, suggesting that *WOX2/3* are essential but not sufficient for SE [69]. In addition, totipotency-related TFs promote SE by modulating auxin biosynthesis, transport, and response. For example, overexpression of *LEC2* in Arabidopsis promoted SE and led to the upregulated expression of auxin biosynthesis related *YUC* genes [70]. Few studies focused on the molecular regulation of SE in woody plants especially roses. Recently, gene expression patterns of two different regenerated materials from cotyledonary somatic embryo were investigated in *R. hybrida* ‘J. F. Kennedy’ by transcriptome analysis [71]. However, there were no more literature on the molecular mechanism for induction and proliferation of SE in roses, which remains us to further explore the in-depth mechanism in the future.

## 4. Materials and Methods

Except for the media for micropropagation have a pH of 6.00 ± 0.05, all of the other media were adjusted to a pH of 5.80 ± 0.05 using 1 mM NaOH or HCl by pH Meter (Sartorius PB-10, Göttingen, Germany), and autoclaved at 121 °C for 20 min (Horizontal Cylindrical Pressure Steam Sterilizer, Jiangyin, China). Filter sterilized TDZ (T8050, Solarbio, Beijing, China), ZT (X8040, Solarbio), GA_3_ (G8040, Solarbio) and ABA (A8060, Solarbio) were added to media when they were cooled to 55–60 °C after sterilization to avoid inactivation. The materials requiring light conditions were cultured at 25 ± 1 °C under a 16 h light/8 h dark photoperiod (2000 lux provided by cool-white fluorescent lights). The materials needed to be cultivated in the dark were covered with black plastic sheeting.

### 4.1. Micropropagation of the Sterile Plantlets

The stems with full and unsprouted lateral buds in the middle of the current-year shoots of cut rose cultivar *R. hybrida* ‘Carola’, grown in the greenhouse from Institute of Vegetables and Flowers, Chinese Academy of Agricultural Science, Beijing, China, was selected as the experimental material. They were divided into two treatments of petiole-removed and non-petiole-removed stems as explants. The stems were rinsed under running water to remove surface dust and then cut into nodal segments with single axillary bud approximately 2 cm in length.

For primary culture, in the ultra-clean bench (Airtech SW-CJ-2FD, Suzhou, China), the nodal segments were immersed with 75% (*v*/*v*) ethanol (Tgreag 64-17-5, Beijing, China) for 1 min and then 0.1% mercuric chloride for 12 min, and finally rinsed at least three times with Sterile Deionized Water (SDW) for 3–5 min. After being dried on sterile paper, each treatment was inoculated with 10 nodal segments in primary media consisting of MS (M8521, Solarbio), 30 g·L^−1^ sucrose (S8271, Solarbio), 6.5 g·L^−1^ carrageenan (C8830, Solarbio) and supplemented with 0.5 mg·L^−1^ 6-BA (A8170, Solarbio), 0.004 mg·L^−1^ NAA (N8010, Solarbio). Each treatment was repeated 3 times. Data were collected on the germination rate, death rate, contamination rate and average height of axillary buds after cultivation for 2 weeks.

For proliferative culture, axillary shoots in 2 cm length with terminal buds of uniform growth were selected and inoculated into the proliferation media, which contained MS, 30 g·L^−1^ sucrose, 6.5 g·L^−1^ carrageenan and supplemented with different concentrations of 6-BA (0.5, 1.0, 1.5 mg·L^−1^) and NAA (0.01, 0.05, 0.1 mg·L^−1^). Each treatment was inoculated with 10 shoots and was replicated 3 times. Data were collected on the ratio of the final number of shoots to the initial number of shoots (proliferation coefficient), average height and growth state of shoots clumps after cultivation for 4 weeks.

For rooting culture, the vigorous shoots with 2 cm in height were excised from shoot clumps and inoculated into the rooting medium, which were composed of 1/2 MS (M8526, Solarbio), 30 g·L^−1^ sucrose, 6.5 g·L^−1^ carrageenan and supplemented with different concentrations of NAA (0–1.0 mg·L^−1^). Each treatment consisted of 3 replicates with 6 shoots. Data were collected on the average number, average length and growth state of roots after cultivation for 3 weeks.

For hardening-off culture, the tissue culture plantlets from rooting medium were placed in the greenhouse (25 ± 2 °C) for different times (3, 6, 9, 12 d), and each treatment was exposed to sunlight for the first half of the day and the sealing film was removed for the second half of the day. The rooted plantlets were removed from culture bottles, and the medium was washed clean from the roots and then transplanted into the mixed substrate with autoclaved vermiculite and peat soil (volume ratio 1:1) covered with plastic film. Each treatment consisted of 3 replicates with 6 rooted plantlets with consistent growth. Data were collected on the survival percentage after cultivation for 3 weeks.

### 4.2. Callus Induction

Unfurled leaflets excised from the in vitro micropropagation plantlets were used for callus induction. The unexpanded compound leaflets of approximately 1 cm were placed abaxial side up on CIM consisting of MS, 30 g·L^−1^ glucose (G8150, Solarbio), 2.5 g·L^−1^ phytagel (P8169, Sigma-Aldrich, St. Louis, MO, USA) and supplemented with different combinations of concentrations of NAA and ZT, KT (K8010, Solarbio), 6-BA. Specific media formulations were prepared according to Table 1. Each treatment consisted of 3 replicates with 10 compound leaves and was incubated in darkness, with the media replaced every 4 weeks. Data were collected on the induction rate of calli and to find the most suitable CIM after cultivation for 8 weeks.

### 4.3. Somatic Embryogenesis

To induce somatic embryos, the calli were incubated for 3 weeks in the most suitable CIM before being transferred to EIM consisting of MS, 30 g·L^−1^ glucose, 2.5 g·L^−1^ phytagel and supplemented with different combinations of concentrations of TDZ, NAA, GA_3_ and ZT, NAA. Specific media formulations were prepared according to Table 2 and Table 3. Each treatment consisted of 3 replicates with 10 calli and was incubated in darkness, with the media replaced every 4 weeks. Data were collected on the induction rate of somatic embryos and to find the most suitable EIM after cultivation for 8 weeks.

### 4.4. Proliferation of Somatic Embryos

Somatic embryos obtained from the most suitable EIM were transferred into EPM consisting of MS, 30 g·L^−1^ glucose, 2.5 g·L^−1^ phytagel and supplemented with different combinations of concentrations of carbohydrate, TDZ and ZT. Specific media formulations were prepared according to Table 4. Each treatment consisted of 3 replicates with 10 somatic embryos and was incubated in light, with the media replaced every 4 weeks. Data were collected on the percentage data of weight of somatic embryos after proliferation to weight of somatic embryos before inoculation (the proliferation coefficient of somatic embryos), the mean value of the percentage of well-established somatic embryos on each somatic embryo (the ratio of somatic embryos) and the growth state of somatic embryos and to find the most suitable EPM after cultivation for 4 weeks.

In order to compare the effect of light conditions on the growth state of somatic embryos, somatic embryos induced from the most suitable EIM were transferred into the most suitable EPM under light or dark conditions. After continuing to culture for 4 weeks, observe the proliferation state of somatic embryos.

### 4.5. Germination of Somatic Embryos

The vigorous somatic embryos in cotyledonary stage were transferred into the EGM consisting of MS, 30 g·L^−1^ glucose, 2.5 g·L^−1^ phytagel and supplemented with different combinations of concentrations of 6-BA, KT, TDZ, GA_3_, ABA and 6-BA, auxins (NAA, IBA (I8030, Solarbio), 2,4-D (ID7260, Solarbio). Specific media formulations were prepared according to Table 5. Each treatment consisted of 3 replicates with 10 somatic embryos and was incubated in light, with the media replaced every 4 weeks. Data were collected on the germination rate of somatic embryos after cultivation for 8 weeks. After continuing to culture for 4 weeks, recorded the number of regenerated plants per somatic embryo and find the optimum EGM.

### 4.6. Rooting and Hardening-Off Culture

Finally, the germinated somatic embryos were transferred into the RM. The culture bottles containing rooted plantlets were placed in the greenhouse (25 ± 2 °C). After 6 days of hardening-off culture, the plantlets were removed from culture bottles, and the medium was washed clean from the roots and then transplanted into the mixed substrate with autoclaved vermiculite and peat soil (volume ratio 1:1) covered.

### 4.7. Statistical Analysis

One-way ANOVA and mean comparisons among treatments by the LSD method (at alpha = 0.05) were performed using SPSS 22 (SPSS, Inc., Chicago, IL, USA). The results were expressed as mean ± standard deviation (SD).

## 5. Conclusions

In this study, an effective in vitro axillary shoot regeneration protocol was firstly established to obtain the sterile and uniform plantlets of *R. hybrida* ‘Carola’. The embryogenic callus was further induced under the suitable medium by using the unexpanded leaflets of sterile plantlets as explants. The optimum medium and culture conditions such as carbohydrate type, PGRs combination etc. were screened for the induction, proliferation and conversion of somatic embryos from the callus-induced indirect pathway. In summary, we established a stable and efficient plant regeneration system through the indirect somatic embryogenesis in the popular cut rose cultivar *R. hybrida* ‘Carola’. The work will provide new protocol for genetic transformation and benefit to the theoretical study on somatic embryogenesis in roses.

## Figures and Tables

**Figure 1 plants-13-03553-f001:**
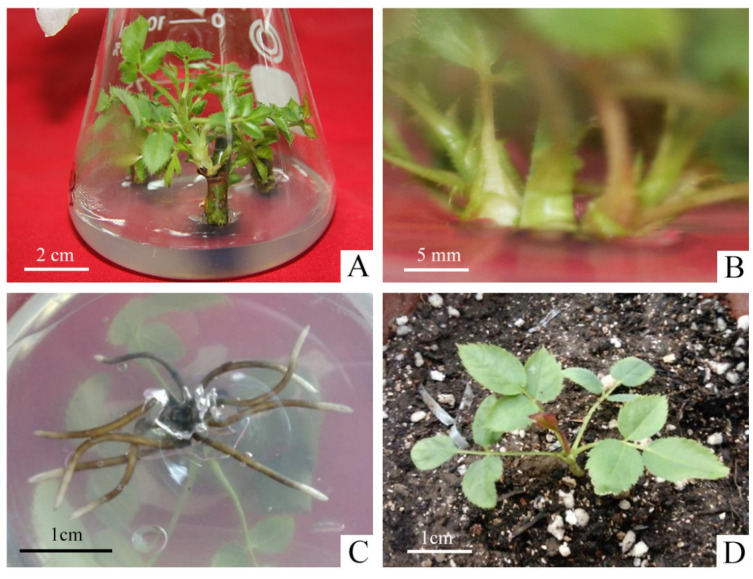
Micropropagation process of *R. hybrida* ‘Carola’. (**A**) Axillary buds produced from nodal segments without petioles in primary medium, (**B**) Proliferation of shoots in proliferation medium, (**C**) Roots produced in rooting medium, (**D**) Rooted plants were transplanted to the greenhouse after hardening-off culture.

**Figure 2 plants-13-03553-f002:**
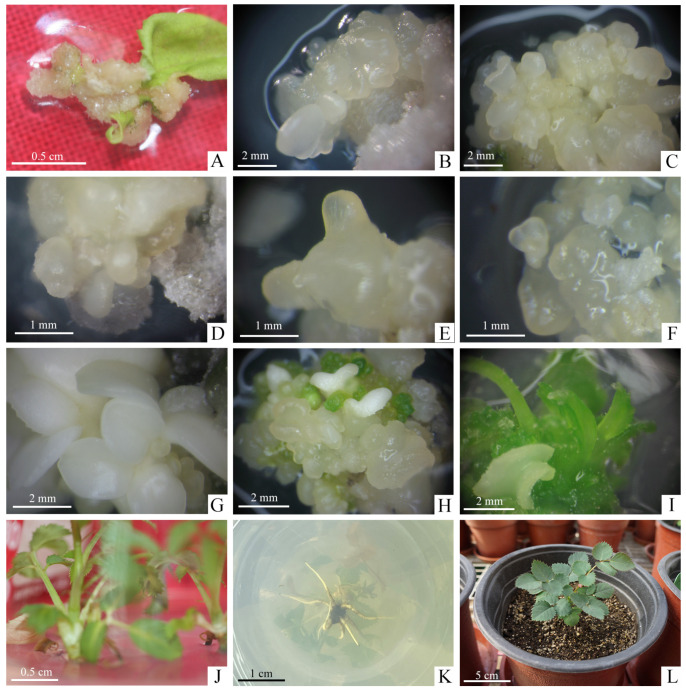
Plant regeneration through somatic embryos in *R. hybrida* ‘Carola’. (**A**) Induction of calli from unexpanded compound leaflet on optimal CIM after cultivation for 4 weeks, (**B**) Induction of EC and somatic embryos initiation on optimal EIM after cultivation for 4 weeks, (**C**) Proliferation of secondary somatic embryos on optimal EPM after cultivation for 4 weeks, (**D**–**G**) Somatic embryos at various stages of development, (**D**) Globular embryos, (**E**) Torpedo embryos, (**F**) Heart-shaped embryos, (**G**) Cotyledonary embryos, (**H**–**J**) Germination process of somatic embryos on optimal EGM during the 8 weeks cultivation, (**H**) Somatic embryos began to turn green, (**I**) Conversion from cotyledons to germinated shoots, (**J**) Stripping of individual shoots from germinated somatic embryo mass, (**K**) Rooting on the RM within 3 weeks, (**L**) Rooted plantlets were transplanted to the greenhouse after hardening-off culture.

**Table 1 plants-13-03553-t001:** Effect of NAA and different concentrations of cytokinins on callus induction after 8 weeks of culture in *R. hybrida* ‘Carola’.

Medium	NAA (mg·L^−1^)	ZT (mg·L^−1^)	KT (mg·L^−1^)	6-BA (mg·L^−1^)	Induction Rate of Callus (%)
A1	2.0	0	0	0	100
A2	2.0	1.0	0	0	100
A3	2.0	0	1.0	0	100
A4	2.0	0	0	1.0	100

**Table 2 plants-13-03553-t002:** Effect of different concentrations of TDZ, NAA and GA_3_ on SE after 8 weeks of culture in *R. hybrida* ‘Carola’.

Medium	TDZ (mg·L^−1^)	NAA (mg·L^−1^)	GA_3_ (mg·L^−1^)	Induction Rate of Somatic Embryos (%)
B1	1.0	0.1	0	0
B2	2.0	0.1	0	6.67 ± 0.07 a
B3	3.0	0.1	0	0
B4	4.0	0.1	0	0
B5	1.0	0.5	0	3.33 ± 0.03 b
B6	2.0	0.5	0	0
B7	3.0	0.5	0	0
B8	4.0	0.5	0	0
B9	1.0	0.5	1.0	10.00 ± 0.1 a
B10	2.0	0.5	1.0	0
B11	3.0	0.5	1.0	0
B12	4.0	0.5	1.0	0

Note: Different lowercase letters in the same column indicate significant differences (*p* < 0.05).

**Table 3 plants-13-03553-t003:** Effect of different concentrations of NAA and ZT on SE after 8 weeks of culture in *R. hybrida* ‘Carola’.

Medium	ZT (mg·L^−1^)	NAA (mg·L^−1^)	Induction Rate of Somatic Embryos (%)
C1	1.0	0.1	0
C2	2.0	0.1	13.33
C3	3.0	0.1	0
C4	4.0	0.1	0
C5	1.0	0.5	0
C6	2.0	0.5	0
C7	3.0	0.5	0
C8	4.0	0.5	0

**Table 4 plants-13-03553-t004:** Effect of different concentrations of carbohydrate and TDZ, ZT on proliferation of somatic embryos after 4 weeks of culture in *R. hybrida* ‘Carola’.

Medium	Glucose (mg·L^−1^)	TDZ (mg·L^−1^)	ZT (mg·L^−1^)	NAA (mg·L^−1^)	GA_3_ (mg·L^−1^)	Proliferation Coefficient	Ratio of Somatic Embryos (%)	Growth State of Somatic Embryos
D1	30	1.5	0	0.2	0.1	2.34 ± 0.13 c	60	Grew weakly with abnormal embryos occur
D2	60	1.5	0	0.2	0.1	3.65 ± 1.56 b	85	Grew vigorously with good gloss
D3	30	0	1.5	0.2	0.1	2.87 ± 0.78 c	72	Grew vigorously with good gloss
D4	60	0	1.5	0.2	0.1	4.02 ± 0.24 a	90	Grew vigorously with good gloss

Note: Different lowercase letters in the same column indicate significant differences (*p* < 0.05).

**Table 5 plants-13-03553-t005:** Effect of different concentrations of 6-BA, KT, GA_3_ and ABA on germination of somatic embryos after 8 or 12 weeks of culture in *R. hybrida* ‘Carola’.

Medium	TDZ (mg·L^−1^)	6-BA (mg·L^−1^)	KT (mg·L^−1^)	GA_3_ (mg·L^−1^)	ABA (mg·L^−1^)	Germination Rate of Somatic Embryos (%)	Number of Regenerated Plantlets per Somatic Embryo
E1	1.5	0.5	0	0	0	17.72 ± 0.10 a	0.80 ± 0.37
E2	1.5	0.5	0	1.0	0	0	0
E3	1.5	0.5	0	0	1.0	0	0
E4	1.5	0	0.5	0	0	0	0

Note: Different lowercase letters in the same column indicate significant differences (*p* < 0.05).

**Table 6 plants-13-03553-t006:** Effect of 6-BA and different kinds of auxins on germination of somatic embryos after 8 or 12 weeks of culture in *R. hybrida* ‘Carola’.

Medium	6-BA (mg·L^−1^)	NAA (mg·L^−1^)	IBA (mg·L^−1^)	2,4-D (mg·L^−1^)	Germination Rate of Somatic Embryos (%)	Number of Regenerated Plantlets per Somatic Embryo
F1	0.5	0	0	0	0	0
F2	1.0	0	0	0	6.90 ± 0.05 b	0.5 ± 0.5 b
F3	1.0	0.01	0	0	10.34 ± 0.07 b	0
F4	1.0	0	0.01	0	43.33 ± 0.24 a	1.31 ± 0.17 a
F5	1.0	0	0	0.01	31.03 ± 0.08 a	0.89 ± 0.20 b

Note: Different lowercase letters in the same column indicate significant differences (*p* < 0.05).

## Data Availability

All data included in the main text.

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
