# Peer review of "Somatic Embryogenesis from the Leaf-Derived Calli of In Vitro Shoot-Regenerated Plantlets of Rosa hybrida ‘Carola’"

_plants, 2024, doi:10.3390/plants13243553_

Round 1
Reviewer 1 Report
Comments and Suggestions for Authors
DearAuthors,
In my opinion, the manuscript presented for the assessment, entitled: "Somatic embryogenesis from the leaf-derived calli of micropropagated seedlings of Rosa hybrida 'Carola'” is generally written in the correct form and the Authors presented a wide range of diverse results. Although the work is very interesting and of application value I think that the Authors should take into account some modification of this article. I recommend publishing it in "Plants” after a minor revision.
General comments: there are many language errors and stylistic inconsistencies in the work, the manuscript should be revised very thoroughly in this respect.
Particular comments:
Title:
“Somatic embryogenesis from the leaf-derived calli of micropropagated seedlings of Rosa hybrida 'Carola'” – does this mean that secondary somatic embryogenesis was induced?
Abstract
Line 5-6: “As an effective method for plant reproduction the regeneration via somatic embryo is the most promising method for breed improvement, factory breeding and genetic …”– plural form of “somatic embryo” will be better
Line 17: “However, lower somatic embryo induction rates and genotypic …. – instead of “somatic embryo” please use “somatic embryogenesis”
Line 32-34: “This study provides the first report of SE in the popular cut rose cultivar R. hybrida 'Carola' and validates the necessity to optimize regeneration protocols for individual cultivars. – should be as follows “… the first report of SE induction …”, and it is not clear from the abstract that this study confirms the need to optimize regeneration protocols for individual strains.
Keywords
Line 35-36: Authors should take into account that keywords, according to the rules of writing scientific papers, should not be the same as in the title and usually in alphabetical order.
Introduction
Line 42: Error! Reference source not found.. ???
Line 54 - 55: “The in vitro regeneration patterns include organ regeneration, somatic embryo
regeneration, protoplast regeneration and protocorm-like body regeneration.” – It should be corrected because the in vitro regeneration patterns are more wide than presented here, please verified it.
Line 57: “…. is applied to many woody plants [13-15]” – not only woody plants, please verified it.
Line 60-61: “…SE has the advantages of bipolarity, independence (independent vascular organization), genetic stability, rapidity, great quantity, and acceptance of foreign DNA, which have great potential for genetic transformation in vitro.” – there is a lack of citations, which should be completed, all the more so because there are reports of disruption of genetic stability of plants obtained through somatic embryogenesis, especially indirect SE e.g. in chrysanthemum cultures and others
Line 132 - 138: This part of the Introduction presents the achievements of the experiments, and there is no clear aim of the work. Please complete it.
Results
Line 189: “CIM” - there is no information what this abbreviation means, I guess, callus inductive medium, but this should have been explained earlier in the text, and similarly with the remaining abbreviations, which should be explained appropriately earlier in the text.
Line 219: “(I) Cotyledons gradually elongated and began to germinate shoots” – it is better to use the term somatic embryos conversion
- usually instead of "the germination somatic embryos" term, the term: somatic embryos conversion" should be better.
Line 286: “2.6. Rooting of somatic embryos” – wrong, the embryos do not take root, they only develop or undergo conversion, which should be corrected.
Discussion
The discussion is long but not exhaustive, lack of in-depth discussion of the influence of the applied growth regulators on the morphogenetic response, their reaction pathways etc...
Materials and Methods
Line 523: “4.6. Rooting and hardening-off culture” – in this part of M&M there is no information about the culture and acclimation condition, which should be completed.
- in M&M there is no information about the equipment on which the culture documentation was prepared, please supplement with complete data on this equipment.
- there is also no information about the source/manufacturer of the chemical reagents (e.g. growth regulators) used in the experiment, which should be completed.
Literature
Lack of the first position citation in the text of the manuscript.
Best regards!
Comments on the Quality of English LanguageDear Authors,
in my opinion, the manuscript's English language is good, but it needs correction.
Reviewer 2 Report
Comments and Suggestions for Authors
Review report
Manuscript ID:plants-3316619
Title: Somatic embryogenesis from the leaf-derived calli of micropropagated seedlings of Rosa hybrida 'Carola'
The authors did not demonstrate the regeneration of adventitious shoot buds. Additionally, they did not explain why explants were taken from in vitro-raised plantlets instead of greenhouse-grown plants for callus induction. They also failed to illustrate the various stages of somatic embryos clearly and somatic embryo germination. Most importantly, they used nonstandard scientific terminology.
Please use standard terms throughout the manuscript.
Title: Please revise the title: The study describes adventitious shoot regeneration and somatic embryogenesis.
Abstract: Please expand the abbreviations.
L16: factory breeding and genetic conservation?
L19: stems with petioles?
L20: 30 g·L-1 glucose? According to materials and methods, it is sucrose (L456 and L462).
L23-24: Please rewrite the sentence.
L30: germinated shoots? A somatic embryo has a bipolar nature, meaning it develops both roots and shoots during the germination process.
L31: 1.0 mg·L-1 NAA. According to the results 2.1 and Table S3, 1.0 mg/L NAA is not suitable for rooting.
L32-34: The authors studied only one cultivar. If they had tested multiple cultivars and identified differences in regeneration ability, they could conclude that it is necessary to optimize regeneration protocols for individual cultivars.
Keywords: Please avoid keywords from the title.
Introduction: Please include a note on the characteristics of Rosa hybrida 'Carola' and its propagation methods.
L42: Error! Reference source not found..?
L53: variety conservation? germplasm conservation.
L54-55: Please rewrite the sentence. plant regeneration, somatic embryogenesis.
L61: genetic stability? Please check the literature.
L76: develop into a plant? convert into a plantlet.
L86: Please expand the abbreviation at the first use (throughout the text).
L109: (GA3) typo.
L111: proliferation? maturation.
L115: Agrobacterium infestation?
L119: number of cotyledons. It represents the final developmental stage of a somatic embryo. (Authors should be aware of the characteristics of both normal and abnormal somatic embryos).
L125: Please delete ‘We also have to be concerned about the number and timing of proliferation.’
L29-136: Please rewrite the objectives clearly. Why did the authors obtain the explants from in vitro-derived plantlets? Adventitious shoot regeneration is more likely to exhibit variations than axillary shoot proliferation.
Results:
L141: with and without petiole?
L142: bud germination? Please edit the manuscript with the subject expert.
L143-144: Please indicate the contamination rate. 26.67% is normal in plant tissue culture.
L146: Please show the contaminated explants. Figure 1A shows axillary shoot development.
L177: Figure 1B does not support adventitious shoot regeneration. Please provide photographs showing different stages of adventitious shoot formation from the explant.
L200: which stage of somatic embryos?
L214: Figure 2: Please specify the type of culture medium used and the age of the culture.
L 217-218: Please provide alternate photographs showing clear stages of somatic embryos: (D) Globular embryos, (E) Torpedo embryos, and (F) Heart-shaped embryos. The current figures do not support this.
L240: structures? stages.
L269: germination and regeneration?
L283: germination and regeneration?
L287: germinated somatic embryos! They have both roots and shoot. Why do we need a rooting medium?
Discussion:
L296-298: Please compare the results with previous studies to identify new findings in the culture medium composition, proliferation coefficient, and survival rate.
L324: (Figure 1B).?
L417: ABA. It is primarily used to prevent asynchronous development of embryos.
L422-425: Table 5 does not support this. In this study, the addition of ABA in the medium resulted in abnormal embryos and significant decreases in the germination rate of somatic embryos, which ultimately no any survived seedling in R. hybrida 'Carola' (Table 5)
L435: Please discuss the results: Rooting of somatic embryos
Materials and Methods:
L448-449: Could you please provide photographs to illustrate the explants for better understanding?
L460: sterile stem sections with terminal buds. Could you please provide photographs to illustrate the explant for better understanding?
L464: Please indicate the number of explants used in each treatment. How did the authors calculate the proliferation coefficient?
L466: aseptic seedlings? shoots
L472: culture seedlings? culture plantlets
L482: Indicate the size of the explant.
L513: Plant germination?
L514: well-grown somatic embryos? Please indicate the stage of somatic embryos.
Conclusions: It should differ from the abstract.
Comments on the Quality of English LanguageThe English could be improved to more clearly express the research.
Round 2
Reviewer 2 Report
Comments and Suggestions for Authors
The manuscript has significantly improved compared to the previous version. However, several typographical errors remain in the manuscript, like "GA3."
L3: Plantlets instead of Seedlings
L206: B5 and B9 media
L235: both media
L236: D1 and D3 media
L260: Please define the abnormal embryos.
L564: axillary shoot
L565: plantlets instead of seedlings
